# Healthcare Impacts Associated with Federally Declared Disasters—Hurricanes Gustave and Ike

**DOI:** 10.3390/ijerph20075388

**Published:** 2023-04-04

**Authors:** Roberta Lavin, Mary Pat Couig, Patricia Watts Kelley, Thais Schwarts, Fermin Ramos

**Affiliations:** 1College of Nursing, University of New Mexico, Albuquerque, NM 87131, USA; 2Clinical and Translational Science Center, University of New Mexico, Albuquerque, NM 87131, USA

**Keywords:** hypertension, diabetes, anxiety, disaster, non-communicable disease

## Abstract

People impacted by disasters may have adverse non-communicable disease health effects associated with the disaster. This research examined the independent and joint impacts of federally declared disasters on the diagnosis of hypertension (HTN), diabetes (DM), anxiety, and medication changes 6 months before and after a disaster. Patients seen in zip codes that received a federal disaster declaration for Hurricanes Gustave or Ike in 2008 and who had electronic health records captured by MarketScan^®^ were analyzed. The analysis included patients seen 6 months before or after Hurricanes Gustav and Ike in 2008 and who were diagnosed with HTN, DM, or anxiety. There was a statistically significant association between post-disaster and diagnosis of hypertension, X^2^ (1, *n* = 19,328) = 3.985, *p* = 0.04. There was no association post-disaster and diabetes X^2^ (1, *n* = 19,328) = 0.778, *p* = 0.378 or anxiety, X^2^ (1, *n* = 19,328) = 0.017, *p* = 0.898. The research showed that there was a change in the diagnosis of HTN after a disaster. Changes in HTN are an additional important consideration for clinicians in disaster-prone areas. Data about non-communicable diseases help healthcare disaster planners to include primary care needs and providers in the plans to prevent the long-term health impacts of disasters and expedite recovery efforts.

## 1. Introduction

The number and frequency of disasters increased by a factor of five in the last 50 years [1]. Disasters are defined as events that can cause human suffering and infrastructure damage [2]. One aspect of human suffering following a disaster is the potential long-term health impacts that often disproportionately affect women and the most marginalized [3]. From the 9/11 air particulates to wildfires to Zika to the Flint water crisis, health impacts endure after the immediate crisis [4,5]. The full extent of the long-term health impacts of COVID-19 is currently unknown, but the evidence is growing that COVID-19 can have long-term health consequences [6]. Furthermore, disasters are inherently stressful events that are part of the typical chronology of the onset of chronic disease after a disaster [7]. While we do not treat diabetes (DM) the same as we treat hypertension (HTN), the National Response Framework uses an all-hazards generalized approach to the disasters [8]. Different health outcomes for different types of disasters would question the approach as it relates to healthcare.

This study addresses gaps in existing research related to new onset metabolic and anxiety disorders following disasters and identifies the impact of the disasters on persons living with these disorders. An emphasis on linking data to social determinants of health (SDOH) and Census data in areas that received federal disaster declarations is included. It was established that the percentage of individuals who live with metabolic and anxiety disorders increased steadily. Among the increased conditions, this study addresses HTN, DM, and anxiety [9].

Like other public health emergencies, African Americans, Latinx, Native Americans, Alaska Natives, and those with low income are disproportionately impacted. An analysis of the Federal Emergency Management Agency (FEMA) National Risk Index Score indicated that U.S. indigenous people are at the greatest risk from natural hazards, including hurricanes, volcanic activity, avalanches, extreme cold, drought, flooding, wildfires, and ice storms. The analysis also showed that Black or African Americans have the lowest overall risk from natural disasters, but they have the highest risk of negative impacts based on expected annual loss, social vulnerability, and community resilience [10,11]. Following Hurricane Katrina, survivors who stayed in a shelter and had a chronic disease requiring medications were twice as likely to present with acute symptoms [12]. Additionally, there is a “differential recognition of medical need as a function of ethnicity” [13]. This raises the question, when linked with the impacts of HTN and DM, is the disaster impact even greater?

The number of people diagnosed with DM also grew from 6.4 million to 26.8 million from 1990 to 2018 [14], with 80% of those diagnosed with DM visiting an emergency department (ED) [15]. Rural patients had higher rates of ED visits than urban patients. Medicare and Medicaid patients were more likely to use EDs for DM-related care [14]. Likewise, more than 40% of adults in rural and 29.4% in urban areas reported HTN, with a 17% higher rate of DM in rural areas. Following Hurricane Sandy, an increase in ER visits by persons with diabetes was found, with the greatest increase by non-Hispanic White individuals [16]. Lee et al. found a statistically significant increase in both primary and secondary diagnosis of DM in ED visits in New York City in the weeks after Hurricane Sandy but did not find changes in the proportion of female, non-Hispanic Black Hispanic, privately or uninsured patients, but there was an increase in Medicaid patients [17]. Most studies that examined chronic disease after disasters found an association with outcomes [7,18,19,20,21].

It was documented that after 9/11, major earthquakes, and Hurricane Katrina, there was an increase in cardiovascular disease and deaths [20]. Two separate review studies of post-disaster elevated BP and DM found that, in most cases, BP returns to normal within six months. However, those with DM were likelier to have sustained HTN [7,20]. Gohardehi et al. found only five U.S. studies that examined disaster survivors specific to HTN and DM, and all focused on 9/11 and Hurricane Katrina [17]. The studies compared disaster survivors to the general population rather than longitudinal studies of individuals diagnosed with HTN or DM before and after a disaster. They consistently found a higher prevalence of HTN and DM among survivors of major disasters than in the general population.

Anxiety disorders are characterized by “unpleasant feelings of stress, uneasiness, tension, and worry [and] is either the predominate disturbance or is expected in confronting a dreaded object or situation or is resisting obsessions or compulsions” [22]. The impact of natural disasters on the development of anxiety disorders was difficult to assess. Many studies used broad definitions of psychological stress rather than specific diagnoses [23,24] or included anxiety, depression, and PTSD [25,26]. Other researchers used large population and Census data for Bayesian modeling to compare mental health outcomes between regions that were impacted and those not impacted by disasters [27,28]. Other studies were reported to oversample people over 60 or required excessive anxiety for 6–9 months [29]. As with HTN and DM, there was no longitudinal study specific to the diagnosis of anxiety before and after a disaster, rather an *a priori* belief that because disasters are inherently stressful events, anxiety would rise.

A frequent concern for people with non-communicable diseases is difficulty obtaining prescribed medications in the aftermath of a disaster. This may be exacerbated by the just-in-time resupply of medications and individuals not having more than a week of medication before the insurance company will pay for a refill. Jimmez-Mangual found that 77% of people in their study of Hurricane Maria survivors reported problems with medications, and 47.7% reported difficulty getting to the pharmacy [30]. In addition to problems with obtaining medications, there are also gender, age, ethnicity, and educational and socio-economic status factors that play a role in inequalities that may exacerbate the universal problem across all disaster types of maintaining medication for chronic conditions [31].

Preliminary studies of mapping capabilities related to disaster risks indicated there is considerable publicly available data on environmental hazards and natural disasters, but a lack of knowledge of the risks and necessary response by healthcare providers [32,33,34]. For example, Veenema et al. [34] identified a full list of U.S. nuclear power sites, including nuclear power reactors and nuclear research and test reactors. [US Nuclear Regulatory Commission]. In that study, the 99 active nuclear power reactors were combined with the five research/test sites that generated the highest amount of power. The sites were layered in ArcGIS with nursing schools, and then, nursing schools were surveyed to determine their perceived risk and related training offered as part of the curriculum. The results showed 75.1% of respondents teach ≤1 h of content on radiation/nuclear care, with 11.5% saying the primary reason for this was a lack of qualified faculty. The faculty (18%) said they would teach it if there were content developed by experts. A total of 53% of respondents answered incorrectly when asked if they were within an emergency planning zone of a nuclear reactor [34].

The United Nations Office on Disaster Risk Reduction (UNDRR) noted the increased risk of disaster resulting from climate change and the growing number of deaths and persons impacted by disasters. “Between 2000 and 2019, 510,837 deaths and 3.9 billion people were affected by 6681 climate-related disasters” [35]. This may further burden the 43.5 million caregivers who provide care to an adult or child in the average year [36]. Caregivers can be either family members or friends who assist a person with health-related task, medications, and medical appointments, usually without compensation [36]. The goal of the research was threefold: (1) to determine the independent and joint impacts of federally declared disasters on the diagnosis and medication changes for a person with HTN, DM, and anxiety six months before and six months after a disaster, (2) link the findings with available FEMA and Census data, and (3) make recommendations for education and practice. Our central hypothesis is that the incidence of long-term metabolic and anxiety disorders following disasters are associated with region-specific conditions disasters and SDOH. This first stage of research involves hurricanes in Louisiana.

Aim 1. Determine the independent and joint impacts of federally declared disasters on metabolic and anxiety disorders 6 months after a disaster. The change will be evaluated to determine the impact of disasters on outcomes for patients with HTN, DM, and/or anxiety. We hypothesize, based on preliminary data, that disaster survivors will show an increased rate of metabolic and anxiety disorders 6 months after the disaster.

Aim 2. Determine the independent and joint impacts of federally declared disasters on the medications prescribed for metabolic and anxiety disorders 6 months after a disaster. The change will be evaluated to determine the impact of disasters on medications. Using the medications prescribed, we hypothesize that disaster survivors diagnosed with metabolic or anxiety disorders have more medications ordered than controls.

Aim 3. Examine regional (local) environmental hazard risk integrating hazard data and data about population vulnerabilities that would affect healthcare workers’ treatment approach. Begin to develop recommendations for provider and community education to mitigate potential disaster health-related risks.

## 2. Materials and Methods

### 2.1. Study Design

This longitudinal, population-based cohort study used electronic health records data reported in MarketScan^®^ (Ann Arbor, MI, USA) from patients seen in zip codes that received a federal disaster declaration because of either Hurricane Gustave or Ike and attempts to answer the proposed research questions. MarketScan^®^ was selected because it has patient-level longitudinal data that maintain the exact date of service. Because pre- and post-disaster data were being compared, the precise dates of service were essential. Additionally, because their data included the lifespan, there was not an overrepresentation of those over 60 years old, which usually occurs with Medicare data.

The conceptual framework of this study was based on the public health crisis conceptual model that was adapted from the crisis conceptual model and the social justice in nursing and public health preparedness human-centered structured analytical approach (Figure 1) [33,37]. The approach was to take the “as is” data specific to the region of the declared disaster in the pre-crisis stage and compare it to the crisis and post-crisis stages. The data and analysis formed the evidence to propose a “to be” educational approach and tools to mitigate the impact on health presented in this article. The process of building an appropriate education component is a continuous cycle of design, development, implementation, evaluation, and analysis that is beyond the scope of this stage of the research.

### 2.2. Data Collection and Sample

In 2008, Hurricane Gustave (Figure 2a) made landfall on 1 September 2008, in Louisiana and Hurricane Ike (Figure 2b) made landfall in Texas on 8 September 2008, with the storm surge impacting Louisiana and causing significant damage. Publicly available FEMA data on federally declared disasters were used to identify areas impacted by the two disasters. Areas were identified by county as having received individual assistance, public assistance, or both.

MarketScan^®^ data comes primarily from large employers; small employers may be underrepresented, and it does not include Medicare (except for those for retirees with employer-paid Medicare supplements). The study included individuals ≥18 who had a documented diagnosis of HTN, DM, or anxiety, with at least one visit to a primary care provider 6 months before and 6 months after a federally declared disaster, were treated in one of the declared regions in Louisiana, and whose healthcare data were captured in MarketScan^®^. As such, the sample was a convenience sample of those whose information is captured in the MarketScan^®^ dataset.

Using the MarketScan^®^ databases of de-identified patient-level health data, ICD-9 codes for HTN, DM, and anxiety were used to identify patients who were diagnosed with HTN, DM, or anxiety 6 months before and after the disasters. Working with the Center for Translational Science Center at the University of New Mexico, SAS Analytical Software was used to collect data from MarketScan^®^. Data were provided to the research team in Excel and imported into SPSS 26 for analysis.

There was a total of 13,681 patients who met the criteria of having a diagnosis of HTN, DM, or anxiety six months before (4 March 2008, to 31 August 2008) or after (14 September to 13 March 2009) Hurricanes Gustave and Ike, and who received care in an area that received a federal disaster designation.

### 2.3. Measures and Analysis

**Noncommunicable chronic conditions** (HTN, DM, anxiety) and medication data were analyzed, and trends were identified, looking for previously unidentified evidence that disaster can cause or exacerbate health conditions. Changes in disease-specific diagnosis and medication use rates were assessed between the pre-disaster and the post-disaster. This is significant because much of the policy related to disaster relief is specific to events caused by a disaster and does not cover events only exacerbated by a disaster.

**Demographic characteristics** of patients were compared using Pearson’s chi-squared test for categorical variables and a *t*-test or Mann–Whitney U test for continuous variables. A patient diagnosis was included if their visit occurred within six months before or six months after the disaster. Individual patients could have more than one diagnosis.

A binary logistic regression was conducted. Association between exposure to disaster (post-disaster) and diagnoses are shown as odds ratios (ORs) with 95% confidence intervals (CIs). No collinearity between variables was observed. The regression models showed an overall statistical significance of α < 0.05 and the goodness-of-fit of α > 0.05 (Hosmer–Lemeshow test). Therefore, the assumptions to conduct the analysis were met. All analyses were conducted using SPSS 26 (IBM, Armonk, NY, USA).

Study 22–215 was submitted to the University of New Mexico, Health Sciences Center IRB, and determined not to be human subjects research.

**Census estimates** for 2008 for Louisiana were used to estimate the total population and population characteristics. FEMA disaster declarations were mapped for the USA from 2012 to 2022 as an estimate of the disaster by region and type. All mapped data were shared on GitHub. Additionally, declared counties were identified and mapped for comparison with the 10-year average.

## 3. Results

### 3.1. Sample Characteristics and Descriptive Results

The overall number of unique patients diagnosed with HTN, DM, and/or anxiety within six months before and after the disaster was 13,681. During the control period before the disaster, 9567 (49.5%) unique patients were diagnosed with HTN, DM, or anxiety, and 9761 (50.5%) after the disaster.

The analysis for medication included a total of 11,911 unique patients (after excluding 92 individuals and 233 medications prescribed due to a diagnosis of HTN, DM, or anxiety not being found). There was no significant difference in medications before or after the disaster in patients diagnosed with HTN, DM, or anxiety.

The patients’ characteristics were similar in the two periods (Table 1). The mean age of the patients before the disaster was 48.3 ± 11.8, with the slightest majority of them being males (50.7%), while the mean age after the disaster was 48.2 ± 11.9 with 50.6% males.

A chi-square test of independence was performed to examine the relationship between the diagnosis of hypertension, diabetes, and anxiety and exposure to the disaster. There was a statistically significant association between post-disaster and diagnosis of hypertension, X^2^ (1, *n* = 19,328) = 3.985, *p* = 0.04. There was no association post-disaster and diabetes X^2^ (1, *n* = 19,328) = 0.778, *p* = 0.378 or anxiety, X^2^ (1, *n* = 19,328) = 0.017, *p* = 0.898.

Binary logistic regression analysis showed that exposure to the disaster was significantly associated with increased odds of being diagnosed with HTN (_a_OR 1.19, 95% CI: 1.09–1.30), DM (_a_OR: 1.13, 95% CI: 1.05–1.23), and anxiety (_a_OR: 1.15, 95% CI: 1.03–1.09) (Table 2) after adjusting for age and sex.

### 3.2. All Medication

Our final analysis for prescribed medication included a total of 11,911 unique patients (after excluding 92 individuals and 233 medications prescribed due to a diagnosis of hypertension, diabetes, or anxiety not being found). A total of 11,911 unique patients were prescribed medications—a total of 8498 unique ones before the disaster and 8671 after the disaster. On average, 3.22 (SD = 2.88) medications per patient were prescribed during the six months preceding the disaster, and 3.21 (SD = 2.87) medications were prescribed after the disaster.

An independent sample *t*-test was conducted to compare the number of medications prescribed for hypertension, diabetes, and anxiety before and after the disaster. Levene’s test (Table 3) for equality of variances was conducted to assess whether the homogeneity of variances assumption was met. The test was found not to violate the assumption of equal variances F(17,167) = 0.039, *p* = 0.844, and a t statistic assuming homogeneity of variance was computed. A statistically significant difference in number of medications for all diagnosis was not found (*t*(17,167) = 0.039, *p* = 0.838; *d* = 0.003) between the two groups. The mean number of medications per patient prescribed before the disaster was 3.22 (SD = 2.88), which was similar to the mean number of medications prescribed after the disaster (M = 3.21, SD = 2.87). The magnitude of the difference in the means was 0.009 (95% CI: −0.077 to 0.095). The effect size (strength of the relationship) for this analysis was (*d* = 0.003), which according to Cohen (1976), is considered a very small effect size.

### 3.3. Hypertension Medication

An independent sample *t*-test was conducted to compare the number of medications prescribed for hypertension before and after the disaster. Levene’s test for equality of variances (Table 4) was conducted to assess whether the homogeneity of variances assumption was met. The test was found not to violate the assumption of equal variances F(8397) = 0.017, *p* = 0.896, and a t statistic assuming homogeneity of variance was computed. A statistically significant difference in number of medications for hypertension was not found (*t*(8397) = 0.017, *p* = 0.386; *d* = 0.019) between the two groups. The mean number of hypertension medications per patient prescribed before the disaster was 2.29 (SD = 1.65), which was similar to the mean number of medications prescribed after the disaster (M = 2.26, SD = 1.67). The magnitude of the difference in the means was 0.031 (95% CI: −0.039 to 0.102).

### 3.4. Diabetes Medication

An independent sample *t*-test was conducted to compare the number of medications prescribed for diabetes before and after the disaster. Levene’s test for equality of variances (Table 5) was conducted to assess whether the homogeneity of variances assumption was met. The test was found not to violate the assumption of equal variances F(3615) = 0.008, *p* = 0.929, and a t statistic assuming homogeneity of variance was computed. A statistically significant difference in number of medications for diabetes was not found (*t*(3615) = −0.094, *p* = 0.925; *d* = −0.003) between the two groups. The mean number of diabetes medications per patient prescribed before the disaster was 4.68 (SD = 3.42), which was similar to the mean number of medications prescribed after the disaster (M = 4.69, SD = 3.47). The magnitude of the difference in the means was −0.011 (95% CI: −0.235 to 0.214).

### 3.5. Anxiety Medications

An independent sample *t*-test was conducted to compare the number of medications prescribed for anxiety before and after the disaster. Levene’s test for equality of variances (Table 6) was conducted to assess whether the homogeneity of variances assumption was met. The test was found not to violate the assumption of equal variances F(5151) = 0.440, *p* = 0.507, and a t statistic assuming homogeneity of variance was computed. A statistically significant difference in the number of medications for diabetes was not found (*t*(5151) = −0.239, *p* = 0.811; *d* = −0.007) between the two groups. The mean number of hypertension medications per patient prescribed before the disaster was 3.70 (SD = 3.45), which was similar to the mean number of medications prescribed after the disaster (M = 3.73, SD = 3.34). The magnitude of the difference in the means was −0.023 (95% CI: −0.208 to 0.163).

Among those diagnosed with hypertension, diabetes, and/or anxiety either before or after the disaster, data were not surprising. In all cases, most people continued their medications. Fewer people diagnosed with any of the conditions discontinued medications than those diagnosed before the disaster and more people diagnosed after the disaster started new medications (Figure 3, Figure 4 and Figure 5).

### 3.6. Census and FEMA Data

The study population consisted of fewer individuals under the age of 55 (62.6%) compared to the Census data on the general population (Table 7) of Louisiana (75.2%) and more individuals over 55 than the general population (37.4% and 24.88%, respectively). However, the study population consisted primarily of working adults. When the 27.6% in the general population that were under the age of 19 were removed, the populations were similar. The study population also had similar numbers of females 18 and over (50.6%) to the general population (51.7%). It should be noted that the study population demographics available from MarketScan^®^ did not include information on race and ethnicity. Therefore, the Census data on race and ethnicity is important.

The United Health Foundation (2008) data analysis ranked Louisiana as 48th for HTN and 44th for DM in the USA. [38]. This indicates that Louisiana is among the states with the highest rates of both non-communicable diseases. There was no information provided on anxiety.

**Table 7 ijerph-20-05388-t007:** 2008 Estimated Louisiana Census Data [39].

	Louisiana			
Label	Estimate	Estimate Margin of Error	Percent	Percent Margin of Error
SEX AND AGE				
Total population	4,544,228	*****	4,544,228	(X)
Male	2,227,014	±4151	49.0%	±0.1
Female	2,317,214	±4151	51.0%	±0.1
Under 5 years	314,008	±2141	6.9%	±0.1
5 to 9 years	311,812	±6977	6.9%	±0.2
10 to 14 years	305,362	±7309	6.7%	±0.2
15 to 19 years	324,812	±4999	7.1%	±0.1
20 to 24 years	342,327	±5712	7.5%	±0.1
25 to 34 years	622,584	±4996	13.7%	±0.1
35 to 44 years	573,691	±5528	12.6%	±0.1
45 to 54 years	649,089	±4549	14.3%	±0.1
55 to 59 years	297,394	±6536	6.5%	±0.1
60 to 64 years	243,560	±6600	5.4%	±0.1
65 to 74 years	313,846	±2497	6.9%	±0.1
75 to 84 years	178,818	±3713	3.9%	±0.1
85 years and over	66,925	±3484	1.5%	±0.1
Median age (years)	35.9	±0.2	(X)	(X)
18 years and over	3,426,176	±2133	75.4%	±0.1
21 years and over	3,216,778	±6780	70.8%	±0.1
62 years and over	699,905	±6492	15.4%	±0.1
65 years and over	559,589	±2415	12.3%	±0.1
18 years and over	3,426,176	±2133	3,426,176	(X)
Male	1,653,635	±2286	48.3%	±0.1
Female	1,772,541	±1844	51.7%	±0.1
65 years and over	559,589	±2415	559,589	(X)
Male	238,072	±1575	42.5%	±0.2
Female	321,517	±1648	57.5%	±0.2
RACE				
Total population	4,544,228	*****	4,544,228	(X)
One race	4,474,462	±5753	98.5%	±0.1
Two or more races	69,766	±5753	1.5%	±0.1
One race	4,474,462	±5753	98.5%	±0.1
White	2,855,924	±5605	62.8%	±0.1
Black or African American	1,455,478	±7095	32.0%	±0.2
American Indian and Alaska Native	28,913	±3426	0.6%	±0.1
Cherokee tribal grouping	2949	±1054	0.1%	±0.1
Chippewa tribal grouping	116	±133	0.0%	±0.1
Navajo tribal grouping	224	±354	0.0%	±0.1
Sioux tribal grouping	319	±259	0.0%	±0.1
Asian	71,476	±3183	1.6%	±0.1
Asian Indian	9931	±3087	0.2%	±0.1
Chinese	10,746	±2252	0.2%	±0.1
Filipino	5205	±1460	0.1%	±0.1
Japanese	2441	±1054	0.1%	±0.1
Korean	2431	±1087	0.1%	±0.1
Vietnamese	28,111	±3821	0.6%	±0.1
Other Asian	12,611	±3478	0.3%	±0.1
Native Hawaiian and Other Pacific Islander	4388	±2907	0.1%	±0.1
Native Hawaiian	1045	±779	0.0%	±0.1
Guamanian or Chamorro	398	±363	0.0%	±0.1
Samoan	407	±388	0.0%	±0.1
Other Pacific Islander	2538	±2658	0.1%	±0.1
Some other race	58,283	±6173	1.3%	±0.1
Two or more races	69,766	±5753	1.5%	±0.1
White and Black or African American	22,709	±4053	0.5%	±0.1
White and American Indian and Alaska Native	17,560	±2575	0.4%	±0.1
White and Asian	7958	±1477	0.2%	±0.1
Black or African American and American Indian and Alaska Native	3497	±1307	0.1%	±0.1
Race alone or in combination with one or more other races				
Total population	4,544,228	*****	4,544,228	(X)
White	2,916,456	±7857	64.2%	±0.2
Black or African American	1,489,050	±6167	32.8%	±0.1
American Indian and Alaska Native	54,590	±3487	1.2%	±0.1
Asian	82,904	±2904	1.8%	±0.1
Native Hawaiian and Other Pacific Islander	8085	±3545	0.2%	±0.1
Some other race	68,906	±5871	1.5%	±0.1
HISPANIC OR LATINO AND RACE				
Total population	4,544,228	*****	4,544,228	(X)
Hispanic or Latino (of any race)	193,732	±1434	4.3%	±0.1
Mexican	82,774	±5211	1.8%	±0.1
Puerto Rican	9370	±1670	0.2%	±0.1
Cuban	11,133	±2506	0.2%	±0.1
Other Hispanic or Latino	90,455	±5479	2.0%	±0.1
Not Hispanic or Latino	4,350,496	±1434	95.7%	±0.1
White alone	2,739,912	±1744	60.3%	±0.1
Black or African American alone	1,446,514	±6904	31.8%	±0.2
American Indian and Alaska Native alone	24,887	±2937	0.5%	±0.1
Asian alone	71,069	±3138	1.6%	±0.1
Native Hawaiian and Other Pacific Islander alone	4261	±2873	0.1%	±0.1
Some other race alone	6692	±2004	0.1%	±0.1
Two or more races	57,161	±5084	1.3%	±0.1
Two races including Some other race	3331	±1116	0.1%	±0.1
Two races excluding Some other race, and Three or more races	53,830	±5013	1.2%	±0.1
Total housing units	1,967,947	±323	(X)	(X)

***** and (X) indicate data that was missing in the U.S. Census data DP05 file. No explanation was provided for the missing data.

The number of hurricanes from the FEMA database is represented in Figure 6 for both Louisiana and the Gulf Coast. The graphic specifically identifies counties with disaster declarations. While all Gulf Coast states except Texas are heavily impacted by hurricanes, Louisiana is the only state that routinely has the entire state impacted heavily by hurricanes, as indicated by the number of counties with disaster declarations.

## 4. Discussion

The first aim explored metabolic and anxiety disorders 6 months after a disaster. The change was evaluated to determine the impact of disasters on outcomes for patients with HTN, DM, and/or anxiety after Hurricanes Gustave and Ike. The number of unique patients diagnosed with HTN, DM, and/or anxiety was similar before (49.5%) and after (50.5%) the disaster. Likewise, the patients’ characteristics were similar in the two periods. The only statistically significant association was the post-disaster diagnosis of hypertension, X^2^ (1, *n* = 19,328) = 3.985, *p* = 0.04. The finding of a significant difference in the post-disaster diagnosis of HTN is inconsistent with the work of Kairo [7] who found that at 6 months, the BP returned to normal except for those that also had DM but consistent with Gohardehi et al. [20], who found a higher incidence of both HTN and DM from disaster survivors. While there were studies that showed increased ED visits for both primary and secondary diagnoses of DM, we did not find studies that showed an increase in the diagnosis of DM, which is consistent with our findings.

The second aim was to determine the independent and joint impacts of federally declared disasters on the medications prescribed for metabolic and anxiety disorders 6 months after a disaster. There was no statistically significant difference in medications before and after the disaster. Still, there was an increase in the number of medications started post-disaster for those diagnosed with HTN, DM, and anxiety. This is consistent with a new diagnosis and not unexpected. The study did not examine difficulties obtaining the medications as our data were related to the prescription.

The third aim examined regional (local) environmental hazard risks integrating hazard data and data about population vulnerabilities that would affect healthcare workers’ treatment approach. The study began a preliminary examination of existing FEMA and Census data to begin to identify interconnections with the post-disaster diagnosis of HTN, DM, and anxiety. The data clearly indicated that Louisiana is among the states most heavily impacted by hurricanes and with the most statewide impact. Because it is 48th in HTN and 44th in DM, their rate of HTN and DM was above the national average before the disaster. While the MarketScan^®^ data did not include information on race and ethnicity, the Census data demonstrated that Louisiana is a highly diverse state, with 32.8% of the population being Black or African American, 1.2% American Indian and Alaska Native, and 1.8% Asian. Because there is a “differential recognition of medical need as a function of ethnicity” [13] it should be expected that healthcare needs following a disaster could vary based on population characteristics and environmental characteristics. Such characteristics should be considered in the precrisis phase planning and decision-making.

We are beginning to develop recommendations for provider education and practice and community education to mitigate potential disaster health-related risks. Clinicians should know the potential hazards in their community and be familiar with community resources to help patients access care, including medications. Recommendations for education and practice include the importance of identifying disaster and health hazards in a region and region specific characteristics. In addition to helping patients prepare for disasters, clinicians must also be alert to potential health hazards resulting from the disaster and aftereffects, e.g., mental health issues, inability to assess medications, and increased risk of non-communicable diseases. With the knowledge that there is a statistically significant association between the six months before and after a disaster and a diagnosis of hypertension, steps can be taken to educate clinicians to screen for hypertension after a disaster and monitor medication compliance, as well as other potential health issues. Additionally, because exposure to the disaster was significantly associated with increased odds of being diagnosed with HTN, DM, and anxiety after adjusting for age and sex, it is recommended that there be increased screening in the primary care setting for these conditions in areas impacted by a disaster.

The Public Health Crisis Conceptual Model (Figure 1) provides a framework for a holistic focus on studying the impact of disasters. Rather than one temporal slice of the disaster, it examines the precrisis, crisis, and postcrisis phases leading to recommendations for education and practice: design, development, implementation, evaluation, and analysis that consider regional disaster and health hazard risks. Using the framework can help ensure local and national data are used, which then facilitates incorporating demographics and potential hazards of the community into provider and student education and training. Emergency planners can use the information in their processes for assessing risk and developing appropriate response and recovery plans. Faculty at health professions schools can incorporate information on the increased risk of HTN, DM, and anxiety into existing content and emphasize the need for added screening following a disaster.

## 5. Limitations

The study was a large retrospective population-based cohort study of people diagnosed with HTN, DM, and/or anxiety 6 months before or after Hurricanes Gustave and Ike and reflected the real-world setting before and after a disaster and served as a model for using the MarketScan^®^ data. MarketScan^®^, unlike many data sets, uses the exact date of services rather than deidentifying information by date; individual patients are coded, making it possible to track a patient before and after a disaster. This is valuable for longitudinal studies that examine health diagnosis and treatment before and after a disaster.The MarketScan^®^ data set was somewhat limited in that it had diagnoses, medications, and tests but did not include all desirable demographic information, such as race and ethnicity, for much of the data.MarketScan^®^ data were obtained primarily from large employers; small employers may be underrepresented, and it did not include Medicare (except for those for retirees with employer-paid Medicare supplements), and as such, not all persons affected by hurricanes Gustave and Ike were included in the data set.MarketScan^®^ is a claims database and, as such, is a convenience sample.

## 6. Conclusions

This research has two important implications for science and practice. First, it was determined that there was a statically significant increase in the diagnosis of HTN but no statically significant changes in the diagnosis of DM, and anxiety after a disaster. It is important to note that there is a key distinction between statistical significance and clinical significance. Although there was no statistically significant increase in the diagnoses of diabetes, anxiety, or prescribed medications, the management of these chronic conditions and the ability to acquire the necessary medication remains critical to the health and well-being of patients, especially during a disaster when people may have been displaced from their homes. Until there is more evidence, it is recommended that increased screening for HTN occur in the 6 months after a disaster. Further studies to evaluate the clinical and statistical significance of different types of disasters in different geographic location on persons living with chronic conditions such as DM, HTN, and anxiety are needed. Further research is ongoing to assess five different types of disasters in five different geographic locations and whether medications were obtained by patients when prescribed.

Second, non-communicable chronic diseases and access to medications are essential considerations for clinicians, educators, and emergency planners in disaster-prone areas. This study illuminated the need to provide guidance and training for healthcare providers, health professions students, and emergency planners ensure that primary care needs, including non-communicable chronic diseases, are addressed in response plans, especially related to long-term health impacts of disasters. The findings related to medications identified a gap in the literature. Specifically, do patients prescribed new medications after a disaster have access to them?

Although the study specifically addressed the impact of natural disasters on HTN, DM, and anxiety, there may be policy implications for implementing primary care screening for metabolic and anxiety disorders following a disaster and changing policy related to what is considered “caused by” a disaster.

## Figures and Tables

**Figure 1 ijerph-20-05388-f001:**
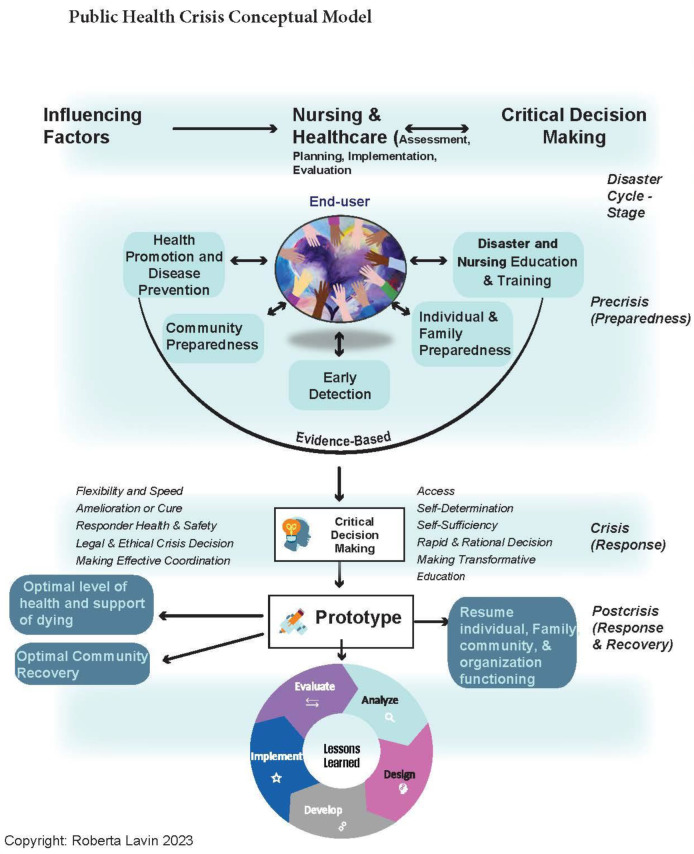
Public Health Crisis Conceptual Model.

**Figure 2 ijerph-20-05388-f002:**
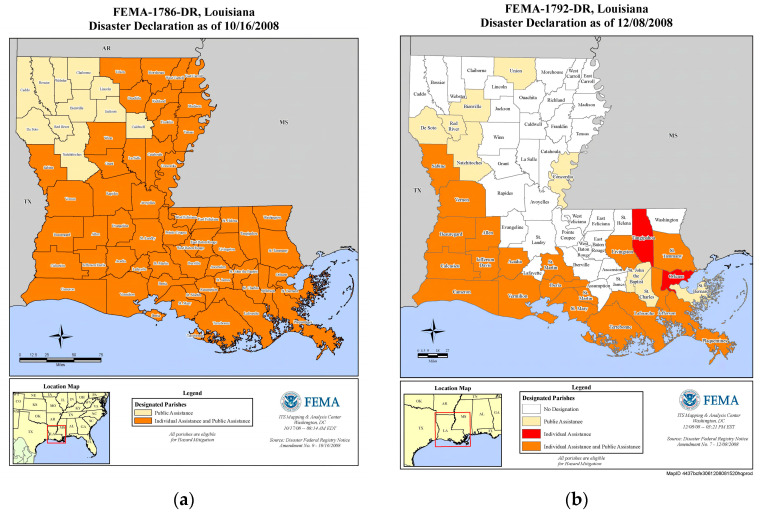
FEMA map of Disaster Assistance by County for Hurricane Gustave and Ike in Louisiana. FEMA produced maps of disaster-declared counties in Louisiana. (**a**) Counties with federal disaster declaration for Hurricane Gustave; (**b**) Counties with federal disaster declarations for hurricane Ike.

**Figure 3 ijerph-20-05388-f003:**
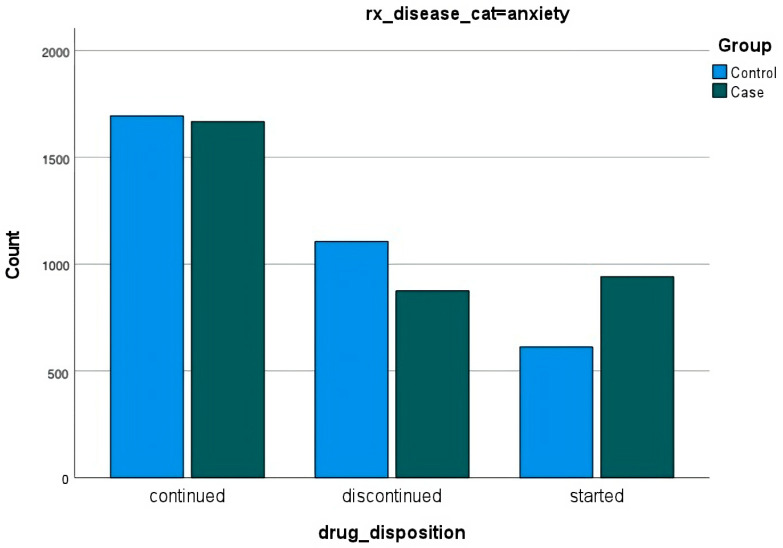
**Prescriptions for anxiety**. Prescriptions for anxiety are represented as continued, discontinued, and started for those diagnosed with anxiety before and after the disaster.

**Figure 4 ijerph-20-05388-f004:**
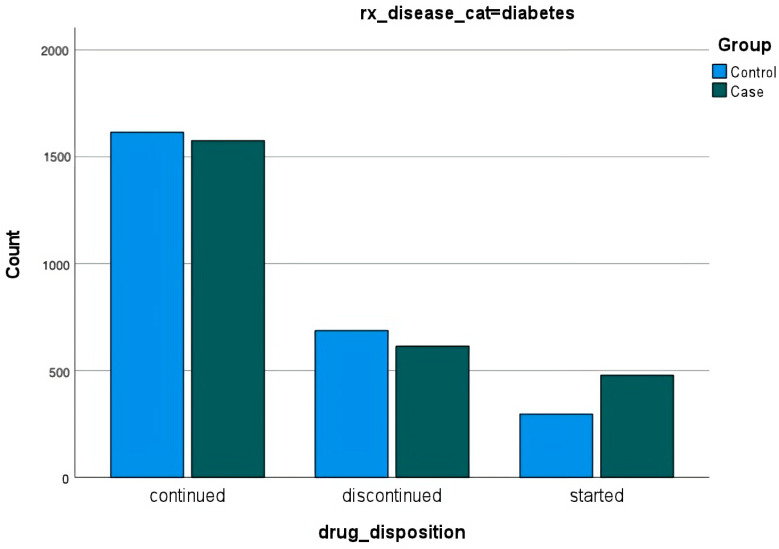
**Prescriptions for diabetes**. Prescriptions for diabetes are represented as continued, discontinued, and started for those diagnosed with anxiety before and after the disaster.

**Figure 5 ijerph-20-05388-f005:**
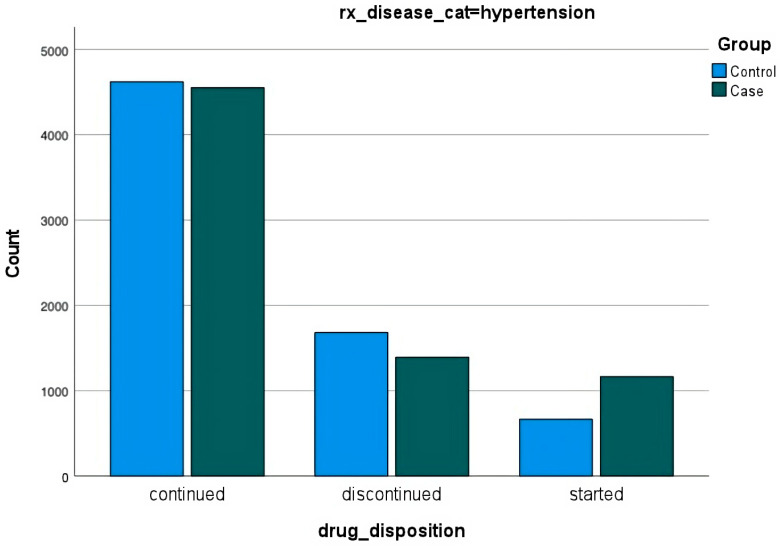
**Prescriptions for hypertension**. Prescriptions for hypertension are represented as continued, discontinued, and started for those diagnosed with anxiety before and after the disaster.

**Figure 6 ijerph-20-05388-f006:**
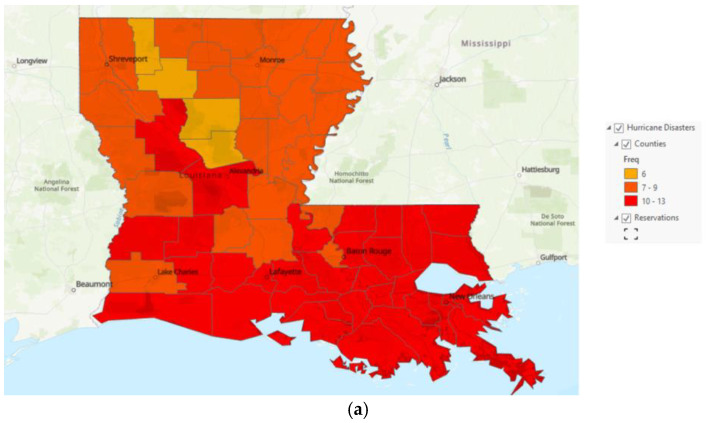
Gulf coast hurricanes are represented for a 10-year period between 2012 and 2022. The greater the number of times impacted, the darker red the color. (**a**) Represents Louisiana only, and (**b**) represents the entire Gulf Coast and Eastern coast of the USA.

**Table 1 ijerph-20-05388-t001:** Characteristics of patients diagnosed with Hypertension, Diabetes, and Anxiety.

	Before Disaster (*n* = 9567)		After Disaster (*n* = 9761)		*p*-Value
Age, years	48.3	11.8	48.2	11.9	0.843
Age Group					
<55 years	5951	62.2	6108	62.6	0.593
≥55 years	3616	37.8	3653	37.4	
Gender					
Male	4852	50.7	4938	50.6	0.86
Female	4715	49.3	4823	49.4	
Diagnosis					
Hypertension	6781	70.9	7045	72.2	0.041
Diabetes	2866	30.0	2981	30.5	0.978
Anxiety	1365	14.3	1399	14.3	0.377
Hypertension Type					
Benign/Malignant	27	0.3	26	0.3	0.358
Benign	2627	27.5	2762	28.3	
Malignant	79	0.8	80	0.8	
Unspecified	4048	42.3	4177	42.8	
Diabetes Type					
Uncontrolled	753	26.3	811	27.2	0.033

**Table 2 ijerph-20-05388-t002:** Odds ratios for the association between exposure to the disaster and diagnosis.

	Diagnosis before Disaster	Diagnosis after Disaster	Unadjusted OR (95% CI)	Adjusted OR (95% CI)	*p*-Value
Hypertension	6781	7045	1.07 (1.00–1.13)	1.19 (1.09–1.30)	<0.001
Diabetes	2866	2981	1.03 (0.97–1.09)	1.13 (1.05–1.23)	0.002
Anxiety	1365	1399	1.01 (0.93–1.09)	1.15 (1.03–1.09)	0.013

OR, Odds ratio; CI, confidence interval.

**Table 3 ijerph-20-05388-t003:** Medications prescribed for HTN, DM, and anxiety.

	Levene’s Test		*t*-Test for Equality of Means						
	F	Sig.	*t*	*df*	Two-Sided *p*	Mean Diff	Std. Error Difference	95% Confidence Interval of the Difference	
								Lower	Upper
Equal variances assumed	0.039	0.844	0.205	17,167	0.838	0.009	0.044	−0.077	0.095
Equal variances not assumed			0.205	17,155.72	0.838	0.009	0.044	−0.077	0.095

**Table 4 ijerph-20-05388-t004:** Independent Samples Test for HTN Medications.

	Levene’s Test		*t*-Test for Equality of Means						
	F	Sig.	*t*	*df*	Two-Sided *p*	Mean Difference	Std. Error Difference	95% Cl of the Difference	
								Lower	Upper
Equal variances assumed	0.017	0.896	0.868	8397	0.386	0.031	0.036	−0.039	0.102
Equal variances not assumed			0.868	8395.612	0.386	0.031	0.036	−0.039	0.102

**Table 5 ijerph-20-05388-t005:** Independent Samples Test for DM Medications.

	Levene’s Test		*t*-Test for Equality of Means						
	F	Sig.	*t*	*df*		Mean Difference	Std. Error Difference	95% Cl of the Difference	
								Lower	Upper
Equal variances assumed	0.008	0.929	−0.094	3615	0.925	−0.011	0.115	−0.235	0.214
Equal variances not assumed			−0.094	3614.671	0.925	−0.011	0.115	−0.235	0.214

**Table 6 ijerph-20-05388-t006:** Independent Samples Test for Anxiety Medications.

	Levene’s Test		*t*-Test for Equality of Means						
	F	Sig.	*t*	*df*		Mean Difference	Std. Error Difference	95% Cl of the Difference	
								Lower	Upper
Equal variances assumed	0.440	0.507	−0.239	5151	0.811	−0.023	0.094	−0.208	0.163
Equal variances not assumed			−0.239	5142.180	0.811	−0.023	0.095	−0.208	0.163

## Data Availability

Restrictions apply to the availability of these data. Data were obtained from MarketScan^®^ and are available from Roberta Lavin with the permission of MarketScan^®^. Mapping data are available on GitHub with permission from Roberta Lavin.

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
