# Peer review of "Healthcare Impacts Associated with Federally Declared Disasters—Hurricanes Gustave and Ike"

_ijerph, 2023, doi:10.3390/ijerph20075388_

Round 1
Reviewer 1 Report
The paper is well written and the study is important. This research examined the independent and joint impacts of federally 9 declared disasters on the diagnosis of hypertension, diabetes, anxiety and medication changes 6 months before and after a disaster.
This is an extremely relevant topic as there is little research on the specific impacts of disasters on health beyond the critical/emergent impacts of the disaster on immediate health and health needs. Researchers endeavor to assess the immediate impact, but the impact on chronic disease is much more difficult to assess. Extant literature assesses emergency room demand, disaster shelter demand and needs, etc. but the literature has a dearth of evidence on the impact of disasters on chronic diseases, and the researchers selected extremely common chronic diseases that impact many.
The methodology is appropriate and well done in its current form. And the conclusions are consistent with the evidence and arguments presented and do they address the main question posed.
I recommend repositioning table 3.
The below correction(s) is/are needed:
Fix this sentence: Data Availability Statement: Restrictions apply to the availability of these data. Data was obtained 366 from MarketScan® and are available Roberta Lavin with the permission of MarketScan®.
There are minor issues with grammar noted (e.g. page 1, line 18 where it states "Changes in HTN is an...." it should read "Changes in HTN are an....". Page 3, line 11, there is a "?" at the end of the sentence, whereas it should be a "."
There may be an issue with the reference software, if that was used, as in several places the sentence starts with a reference number instead of the author name. For example, page 2 line 52, the sentence is "11 found....." and in line 61 on the same page is says "17 found" and finally on the same page in line 78 it states "24 found"
On page 4, line 155, there should be a period at the end of the sentence.
These are all minor corrections, and the rest of the paper is excellent.
Author Response
I would like to thank all of the reviewers for their work on this article. I think their recommendations significantly improved the quality and readability of the article. I tried to address each comment. The document is included in track changes and I included a copy with all the corrections accepted for readability.
Reviewer 1
- I recommend repositioning table 3.
- I wasn’t sure where they would want this repositioned to so we left it as is.
- Fix this sentence: Data Availability Statement: Restrictions apply to the availability of these data. Data was obtained 366 from MarketScan® and are available Roberta Lavin with the permission of MarketScan®.
- Done
- There are minor issues with grammar noted (e.g. page 1, line 18 where it states "Changes in HTN is an...." it should read "Changes in HTN are an....". Page 3, line 11, there is a "?" at the end of the sentence, whereas it should be a "."
- Corrected
- There may be an issue with the reference software, if that was used, as in several places the sentence starts with a reference number instead of the author name. For example, page 2 line 52, the sentence is "11 found....." and in line 61 on the same page is says "17 found" and finally on the same page in line 78 it states "24 found"
- This was a reference manager issue and all have been fixed.
- On page 4, line 155, there should be a period at the end of the sentence.
- Fixed

Reviewer 2 Report
First of all, although it is not a new theme, it is one that needs intervention and change, so I consider it very pertinent and necessary.
The study needs improvement, so I leave some suggestions:
In the introduction it is necessary to better frame the theme. It should be made a better presentation of the disease/diagnosis, I suggest the DSM-V as one of the sources. This is to understand the impact of the disease on the person and later on the family/caregivers.
The definition of carers should also be included. As well as data on family caregiver burden should also be addressed, either nationally and/or internationally.
In "2. Materials and Methods" the design of the study should be initially set out (nature and type of study, and theoretical rationale for the choice in confrontation with the aim of the study).
In this section it is also important to indicate objectively the sampling method, justification and rationale of the choice.
The conclusion does not reflect the overall results of the study and should be further explored.
The implications of these results for the care settings could also be further explored with more suggestions for improving interventions/accompaniment.
Author Response
- In the introduction it is necessary to better frame the theme. It should be made a better presentation of the disease/diagnosis, I suggest the DSM-V as one of the sources. This is to understand the impact of the disease on the person and later on the family/caregivers.
- We added a definition of anxiety though we chose to use the definition from Corsini as it is easier for a non-psychologist to understand and is more consistent with the research, which depended on ICD-9 codes.
- The definition of carers should also be included. As well as data on family caregiver burden should also be addressed, either nationally and/or internationally.
- We added this definition, but the research really was not related to carers or caregiver burden. There is no Aim or hypothesis related to carer burden. While I think this would be interesting research, it was not part of our study. However, we did attempt to address the comment.
- In "2. Materials and Methods" the design of the study should be initially set out (nature and type of study, and theoretical rationale for the choice in confrontation with the aim of the study).
- This section was significantly revised with much more detail.
- In this section it is also important to indicate objectively the sampling method, justification and rationale of the choice.
- The conclusion does not reflect the overall results of the study and should be further explored.
- We added more detail to the conclusion and to the discussion.
- The implications of these results for the care settings could also be further explored with more suggestions for improving interventions/accompaniment.
- HTN, DM, and anxiety are primarily treated in primary care. We are not sure what is meant by accompaniment. The population included is primarily working adults with a first diagnosis of HTN, DM, or anxiety.
- We did make suggestions for education and planning, but treatment recommendations are outside the scope of the study except for increased screening following a disaster. We did make some additional recommendations related to screening and training.

Reviewer 3 Report
<Introduction>
-line 43: You need scientific evidence to claim that 'the impact is much greater when linked to that of HTN and DM'. Please provide a reference.
-line45-47: What is the correlation between the increase in natural disasters and the increase in DM diagnoses? Simplified explanations are inappropriate as it entails a quantitative increase.
-I think the purpose of this study is too broad. Is it a study on the relationship between HTN and DM, disasters and anxiety caused by disasters? Is it a research study to nurture nursing education courses and education experts who can respond to disasters and disasters? For research purposes, it should be clearer and more focused.
-line. 52: Wouldn't it be correct to capitalize the F in found?
<Discussion>
The purpose of the study is '3) Do you make suggestions for education and practice', but there is no discussion on this.
Author Response
I would like to thank all of the reviewers for their work on this article. I think their recommendations significantly improved the quality and readability of the article. I tried to address each comment. The document is included in track changes and I included a copy with all the corrections accepted for readability.
Reviewer 3
- line 43: You need scientific evidence to claim that 'the impact is much greater when linked to that of HTN and DM'. Please provide a reference.
- This has been corrected and rephrased.
- Line 45-47: What is the correlation between the increase in natural disasters and the increase in DM diagnoses? Simplified explanations are inappropriate as it entails a quantitative increase.
- This was not a study of the correlation between the increased incidence of disasters and an increase in DM diagnosis, but we pointed out in the literature review that both were increasing, lest readers say we were not aware of the increase in both. However, our study looked at individual patient data before and after the disaster thus, even if both were increasing, the study design controlled for it. The study focuses on the independent and joint impacts of federally declared disasters on metabolic and anxiety disorders 6 months after a disaster. The change was evaluated to determine the impact of disasters on outcomes for patients with HTN, DM, and/or anxiety. We hypothesized, based on preliminary data, that disaster survivors will show an increased rate of metabolic and anxiety disorders 6 months after the disaster. As was shown in the data we did not find a statistically significant difference in the incidence of DM, but the odds were increased when adjusting for age.
- I think the purpose of this study is too broad. Is it a study on the relationship between HTN and DM, disasters and anxiety caused by disasters? Is it a research study to nurture nursing education courses and education experts who can respond to disasters and disasters? For research purposes, it should be clearer and more focused.
- This study was not related to nursing education. I think because I’m a nurse the article is included in nursing, but the study was about the impact of disasters on the diagnosis of HTN, DM, and anxiety and the prescribed medication. I added the aims to the study which I think clarifies the purpose.
- 52: Wouldn't it be correct to capitalize the F in found?
- This was a problem with the reference manager which has now been corrected.
- <Discussion> The purpose of the study is '3) Do you make suggestions for education and practice', but there is no discussion on this.
- We have clarified this in the discussion.

Round 2
Reviewer 2 Report
The authors responded to all suggestions for improvement improving the quality of the article for publication.